# Antibacterial Activity of Two Zn-MOFs Containing a Tricarboxylate Linker

**DOI:** 10.3390/nano12234139

**Published:** 2022-11-23

**Authors:** Sara Rojas, Amalia García-García, Tania Hidalgo, María Rosales, Daniel Ruiz-Camino, Pablo Salcedo-Abraira, Helena Montes-Andrés, Duane Choquesillo-Lazarte, Roberto Rosal, Patricia Horcajada, Antonio Rodríguez-Diéguez

**Affiliations:** 1Department of Inorganic Chemistry, Faculty of Science, University of Granada, Av. Fuentenueva s/n, 18071 Granada, Spain; 2Advanced Porous Materials Unit, IMDEA Energy Institute, Av. Ramón de la Sagra 3, 28935 Móstoles, Spain; 3Laboratorio de Estudios Cristalográficos, IACT-CSIC, Av. de las Palmeras, 4, 18100 Armilla, Spain; 4Department of Chemical Engineering, University of Alcalá, 28871 Alcalá de Henares, Spain

**Keywords:** metal–organic frameworks, zinc, antibacterial activity

## Abstract

Metal–organic frameworks (MOFs) can be used as reservoirs of metal ions with relevant antibacterial effects. Here, two novel Zn-based MOFs with the formulas [Zn_4_(μ_4_-O)(μ-FA)L_2_] (GR-MOF-8) and [Zn_4_(μ_4_-O)L_2_(H_2_O)] (GR-MOF-9) (H_3_L: 5-((4-carboxyphenyl)ethynyl) in isophthalic acid and FA (formate anion) were solvothermally synthetized and fully characterized. The antibacterial activity of GR-MOF-8 and 9 was investigated against *Staphylococcus aureus* (SA) and *Escherichia Coli* (*EC*) by the agar diffusion method. Both bacteria are among the most relevant human and animal pathogens, causing a wide variety of infections, and are often related with the development of antimicrobial resistances. While both Zn-based materials exhibited antibacterial activity against both strains, GR-MOF-8 showed the highest inhibitory action, likely due to a more progressive Zn release under the tested experimental conditions. This is particularly evidenced in the inhibition of *SA*, with an increasing effect of GR-MOF-8 with time, which is of great significance to ensure the disappearance of the microorganism.

## 1. Introduction

The spread of drug-resistant bacteria continues to threaten our ability to treat common infections due to their growing rate of new resistance mechanisms. According to the World Health Organization (WHO), the rapid global proliferation of multi- and pan-resistant bacteria (also known as “superbugs”) that cause non-curable infections with the existing antimicrobial medicines such as antibiotics, antivirals, antifungals, etc., is especially alarming. [1]. Bearing in mind that the newest antibiotic class was introduced in 2003, the treatment of many infectious diseases remains a challenge [2]. According to the Antimicrobial Resistance of the European Union (EU) and European Economic Area (EEA) 2020 report, each year more than 670 thousand infections occur in the EU/EEA due to bacteria resistant to antibiotics, and that approximately 33 thousand people die as a direct consequence of these infections [3]. The related cost to the healthcare systems of EU/EEA countries is estimated to be around EUR 1.1 billion *per* year. Different strategies have been proposed to control the development of antibiotic resistance, such as control the prescription of antibiotics, or control the already resistant bacteria, like the development of novel antibacterial agents. In this line, new strategies are urgently required to address the associated issues of the post-antibiotic era.

Various transition metals (Zn, Ag, Cu, etc.) and metallic nanoparticles (ZnO, Au, etc.) [4] have gained prominence as substitutes for conventional antibacterial agents. Zinc is an essential element for the human body, involved in many biological pathways and metabolic activities, such as an enzymatic cofactor, signaling molecule or as a structural element [5]. Zn has an antioxidant and anti-inflammatory effect, associated to: (*i*) the potential activation of the antioxidant enzyme superoxide dismutase (SOD1 and 3), which possesses Zn and Cu in its active metal site, and (*ii*) the inhibition of the nicotinamide adenine dinucleotide phosphate (NADPH) oxidase, involved in the free radicals production [6,7]. Furthermore, Zn-based nanoparticles have shown excellent antibacterial properties, although high doses of Zn^2+^ can be harmful to normal tissues or even lead to the development of bacteria resistances. In this regard, metal–organic frameworks (MOFs) have been studied as antibacterial materials, acting as reservoirs of metal ions [8]. MOFs are considered as a promising family of porous coordination polymers, comprising inorganic nodes (*e.g.,* atoms, clusters, chains) and organic polycomplexant linkers (*e.g.,* carboxylates, phosphonates, azolates) that assemble into multidimensional periodic lattices [9]. These materials have been proposed for diverse applications including adsorption [10], separation [11], sensing [12], catalysis [13], fuel cells [14], drug delivery [15], light harvesting [16], photocatalysis [17], biocatalysis [18], etc. In particular, Zn-based MOFs can be seen as promising reservoir materials of metal ions with antimicrobial activity [19]. However, so far, only a few examples of Zn-based MOFs with antibacterial activity have been reported. Wang et al. reported a series of four MOFs based on Zn^2+^ and 2,2′-bipyridine-4,4′-dicarboxylic acid with different morphologies and architectures (1D nanobelts, 3D nanorod-flowers, fan-like and rhombus-like) [20]. The antimicrobial studies against *Bacillus subtilis, Staphylococcus aureus, Salmonella enteritidis, Escherichia coli, Proteus vulgaris* and *Pseudomonas aeruginosa* demonstrated the influence of the MOF particles’ morphology on their activity, particularly that fan-like particles are more biocides against Gram-negative bacteria than other morphologies. A further study reported the possible preparation of a hydrogel containing a previously reported Zn-based MOFs with antibacterial activity against *S. aureus* and *E. coli* [21]. Some of us described the MOF material BioMIL-5, based on Zn^2+^ and the natural dicarboxylic acid, azelaic acid [22]. When tested against *S. aureus* and *Staphylococcus epidermidis*, BioMIL-5 demonstrated that the antimicrobial activity of the individual components of BioMIL-5 were maintained after its synthesis. Another recent example is the effect of the media in the antibacterial activity of the Zn-based MOF ZIF-8, with superior antibacterial activity against *E. coli* in phosphate-buffered saline (PBS) of Luria Bertani (LB) compared with pure water [23]. 

Following this trend, in the present study we report the solvothermal synthesis of two novel MOF structures based on Zn^2+^ as an inorganic node and the tricarboxylate ligand 5-((4-carboxyphenyl)ethynyl) isophthalic acid (H_3_L; linker recently reported by some of us [24]), named GR-MOF-8 and 9, both with interesting antibacterial properties. First, their structures were determined by single crystal X-ray diffraction (SCXRD) and fully characterized following different physicochemical characterization techniques (Fourier transform infrared (FTIR), elemental and thermogravimetry analyses, etc.). Finally, their antibacterial performance was evaluated by using two bacteria, representatives of Gram-positive and -negative strains: *Staphylococcus aureus (SA)* and *Escherichia coli (EC)*, respectively.

## 2. Materials and Methods

Chemicals were readily available from commercial sources (Sigma-Aldrich, Merck Group, Darmstadt, Germany) and used as received without further purification.


**Synthesis of 5-((4-carboxyphenyl)ethynyl) isophthalic acid (H_3_L)**


The procedure used in the synthesis of H_3_L linker was the same as previously described [24].


**Synthesis of GR-MOF-8**


A total of 0.5 mL of a *N*,*N*-dimethylformamide (DMF) solution containing 0.12 mmol (21 mg) of Zn(AcO)_2_·2H_2_O was added dropwise under continuous stirring over 0.5 mL of an ethanol (EtOH) solution containing 0.03 mmol (10 mg) of H_3_L at 80 °C. The final solution was introduced in a sealed vial, where it was heated up to 100 °C in an oven for 24 h. The mixture was slowly cooled down to room temperature and clear foil shape X-ray quality crystals were obtained. The obtained solid was filtered off and washed several times with DMF. Yield based on metal: 27%. Proposed formula: [Zn_4_(μ_4_-O)(μ-CO_2_)(C_17_O_6_H_7_)_2_] or [Zn_4_(μ_4_-O)(μ-FA)L_2_]. MW: 935.94 g·mol^−1^. Elemental analysis (weight, %); Calculated: C, 47.15; H, 3.22; N, 3.84; Found: C, 42.12; H, 4.85; N, 3.28.


**Synthesis of GR-MOF-9**


A total of 0.5 mL of a DMF solution containing 0.06 mmol (0.11 mg) of Zn(AcO)_2_·2H_2_O was added dropwise under continuous stirring over 0.5 mL of a DMF solution containing 0.03 mmol (10 mg) of H_3_L at 80 °C. The final solution was introduced in a sealed vial, where it was heated up to 100 °C in an oven for 24 h. After this time, octahedral shaped X-ray quality crystals were obtained. Yield based on metal: 21%. Proposed formula: [Zn_4_(μ_4_-O)(C_17_O_6_H_7_)_2_(H_2_O)] or [Zn_4_(μ_4_-O)L_2_(H_2_O)]. MW: 909.95 g·mol^−1^. Elemental analysis (weight, %); Calculated: C, 47.15; H, 3.22; N, 3.84; Found: C, 47.15; H, 3.22; N, 3.84.

### 2.1. Physicochemical Characterization

Elemental analyses were carried out on a THERMO SCIENTIFIC analyzer model Flash 2000. The Fourier transform infrared (FTIR) spectra measured on powdered samples were recorded on a BRUKER TENSOR 27 FT-IR and OPUS data collection program. Powder X-ray diffraction (PXRD) patterns were registered in a BRUKER D8 ADVANCE equipment. The routine PXRD conditions were from 3 to 35° (2*θ*) using a step size of 0.013° and 39525 s *per* step in continuous mode with knife and with Soller slits of 0.04 rad. The same conditions were used for the Le Bail fitting. Nitrogen isotherms were obtained at 77 K using an AutosorbQ2 (Quantachrome Instruments, Illinois, USA). Before the measurements, all samples were evacuated at 170 °C overnight under vacuum. Thermogravimetric analyses (TGA) were carried out in a SDT A-600 thermobalance (TA Instruments, New Castle, DE, USA) with a general heating profile from 30 to 600 °C with a heating rate of 5 °C·min^−1^. Nuclear magnetic resonance (NMR) spectra were measured in a Bruker Advance III NEO 400 MHz spectrometer equipped with a direct double Smart iProbe 5 mm with Z gradient. Variable-temperature powder X-ray diffraction (VTPXRD) data were collected on a D8 Advance Bruker AXS *θ*-2*θ* diffractometer (Cu *Kα* X-radiation, λ = 1.5418 Å), equipped with a LYNXEYE XE detector, operating at 40 kV and 40 mA and an Anton Parr XRK 900 high-temperature chamber. VTPXRD intensity data were collected under a continuous compressed air flow (10 mL·min^−1^) in the step mode (0.01° 2*θ*, 0.1 s *per* step) in the range of 3–30° 2*θ* between 25 and 500 °C. The heating ramp used was 5 °C·min^−1^.

### 2.2. Antibacterial Activity

The Gram-positive *SA* (CECT 240, strain designation ATCC 6538P) and Gram-negative *EC* (CECT 516, strain designation ATCC 8739) bacteria were used as reference strains for the antibacterial testing. The microorganisms were preserved at −80 °C in glycerol (20% *v*/*v*) until their use. Reactivation was performed in nutrient broth (1 mL of inoculum in 20 mL of NB; composed of 5 g·L^−1^ beef extract, 10 g·L^−1^ peptone, 5 g·L^−1^ NaCl, pH = 7–7.2) at 37 °C under stirring (100 rpm) and routinely tracked by measuring OD at 600 nm (Shimadzu UV-1800 spectrophotometer) to preserve the exponentially growing phase of the microorganisms.

The bacterial inhibition effect of the tested materials was evaluated using the solid agar diffusion method: 5 mg of each GR-MOF-8 or 9 along with the corresponding percentage of each precursor (the organic ligand H_3_L (49.5%) and metallic salt Zn(AcO)_2_·H_2_O (*ca.* 10.5%)) were placed on the surface of soft agar plates, previously inoculated with 0.6 mL of the specific bacterial suspension (~10^6^ colony forming units (CFUs) *per* mL of *SA* and *EC*). Plates were incubated at 37 °C for 14 days. At specific time intervals, the plates were digitally photographed, and the diameter of the inhibition zone recorded as a measurement of the antibacterial activity compared to the negative controls (each bacterial suspension onto the solid agar plate with regular strain growth—100% viability). All experiments were repeated at least three times.

### 2.3. Crystallographic Refinement and Structure Solution

Single crystals of suitable dimensions of GR-MOF-8 and 9 were used for data collection. Diffraction intensities were recorded on a Bruker APEX II CCD and Bruker D8 Venture with a Photon detector equipped with graphite monochromated MoKα radiation (λ = 0.71073 Å). The data reduction was performed with the APEX4 (v2021.1, Bruker, Madison, USA,) software [25], and corrected for absorption using SADABS [26]. In both cases, the structures were solved by direct methods and refined by full-matrix least-squares with SHELXL [27]. The OLEX2-1.5 software was utilized as the graphical interface [28]. During the structure refinement of GR-MOF-8, regions of electron density were identified as highly disordered DMF molecules. Attempts to model these electron densities as water were not successful due to the extent of the disorder. In the final structure model, the contribution of the electron density from 1.25 DMF molecules *per* formula unit has been removed from the intensity data using the solvent mask tool in Olex2. SADI, ISOR and RIGU commands were used to obtain reasonable geometry and anisotropic displacement parameters for the ligands. The main refinement parameters are listed in Table 1. Details of selected bond, lengths and angles are given in Appendix A. CCDC reference number is 2207753 for GR-MOF-9 material.

## 3. Results and Discussion

### 3.1. Synthesis and Crystal Structure Description of GR-MOF-8 and 9

Two novel MOFs (named GR-MOF-8 and 9), based on Zn^2+^ and the tricarboxylate linker H_3_L, were successfully isolated upon exhaustive optimization of reaction conditions. Briefly, a mixture 50:50 of *N*,*N*-dimethylformamide (DMF) and ethanol (EtOH) solution for GR-MOF-8 and only DMF for GR-MOF-9 composed of Zn(AcO)_2_·2H_2_O and H_3_L were heated at 100 °C for 24 h (see Experimental Section for further details), leading to two 3D structures, formulated as [Zn_4_(μ_4_-O)(μ-FA)L_2_] for GR-MOF-8, where FA is formate anion (coming from the degradation of the DMF solvent), and as [Zn_4_(μ_4_-O)L_2_(H_2_O)] for GR-MOF-9 (Figure 1).

Both materials were prepared in high purity, suitable for their structure solution by single-crystal X-ray diffraction (SCXRD). As revealed by SCXRD, GR-MOF-8 crystallizes in the monoclinic space group *P2_1_*/*c* and consists of an open 3D structure with voids along the *a* crystallographic axis. The asymmetric unit contains four Zn^II^ atoms, two ligand molecules, one formate molecule and one hydroxide anion, as it is reflected by its formula. The metal coordination node consists of [Zn_4_O_13_] tetrahedra sharing the same oxygen atom (O24) which is the center of the structure (Figure 2). The Zn–O bond distances in the range 1.910(10)-2.011(11) Å are standard for similar compounds [29,30]. Metal centers and their coordination with the linkers lead to the formation of the pores of the structure, with dimensions estimated as 18.0 × 10.9 Å.

A similar type of cluster is observed in GR-MOF-9, with a [Zn_4_O_14_] core, but this time a more complex structure is crystallized, namely the tetragonal space group *I-4*. Its asymmetric unit is made up of four Zn^II^ atoms, two ligand molecules, one water molecule and an oxo bridging atom. In a similar way, the Zn–O bond lengths are between 1.92(7) and 2.33(6) Å.

The observed complexity of these structures could be related with the diverse coordination modes of the linker (Appendix A), since it possesses three carboxylate groups. In the case of GR-MOF-8, two non-equivalent linkers are shown: one coordinated to six Zn^II^ atoms through all its oxygen atoms, whereas the other coordinates only to four Zn^II^ atoms by the monodentate mode. This disposition creates rectangular channels along the *a* axis, in which formate anions point inward. On the other hand, GR-MOF-9 possesses each linker coordinated to six Zn^II^ atoms, which results in a more complex structure with triangle channels (*ca.* 18 Å) along the *a* axis.

### 3.2. Physicochemical Characterization

The purity of the obtained polycrystalline new MOFs was confirmed by a Le Bail profile fitting using powder X-ray diffraction (PXRD, Appendix A). Furthermore, when compared with the free H_3_L linker, the FTIR spectra of GR-MOF-8 and 9 (Appendix A) show the displacement to lower frequencies of the two carboxylic bands (from ca. 1745 and 1679 cm^−1^ to ca. 1654 and 1603 cm^−1^ for GR-MOF-8 and 9, respectively), indicating the absence of a free linker in the powdered materials. The chemical composition of GR-MOF-8 and 9 was also confirmed by elemental analysis (C, H and N): Theo.(weight, %): C 47.15, H 3.22, N 3.84; Exp.: C 42.12, H 4.85, N 3.28 (GR-MOF-8); and Theo.(weight, %): C 47.15, H 3.22, N 3.8;. Exp.: C 47.15, H 3.22, N 3.84 (GR-MOF-9).

The thermogravimetric analysis (TGA) of GR-MOF-8 and 9 is slightly similar, showing a steed with a progressive weight loss later on (from room temperature (RT) to 150 °C first and from 150 to 250 °C, afterward), attributed to the DMF chemi- and physiosorbed on the internal and external surface of the crystals (Appendix A). The following mass loss (starting at 370 °C) could correspond to the MOF decomposition, associated with the organic ligand oxidation. The thermal stability of both materials was more precisely evaluated by variable-temperature powder X-ray diffraction (VTPXRD, Appendix A). GR-MOF-8 and 9 materials are stable up to 370 and 350 °C, respectively, leading then to an amorphous phase. In the case of GR-MOF-8, a phase transition is observed at around 100 to 150 °C, which may be associated with the departure of crystallization DMF molecules, in agreement with the TGA data.

Considering the potential accessible porosity of GR-MOF-9 (67 vol.% free *per* unit cell, estimated by Platon), gas sorption experiments were also performed using N_2_ as an adsorbate model. In this case, after an overnight activation at different temperatures (from 90 to 170 °C), a minimal Brunauer, Emmett and Teller surface area (S_BET_) value is obtained (90 m^2^·g^−1^ at 170 ºC), indicating that GR-MOF-9 shows no accessible porosity to N_2_ at 77 K.

### 3.3. Antibacterial Activity

Considering the recognized antibacterial activity of Zn^2+^ and the potential of MOFs in the progressive release of metals, the potential antimicrobial properties of these novel resulting materials was tested against *Staphylococcus aureus* (*SA*; *Gram*(+)) and *Escherichia coli* (*EC*; *Gram*(-)) strains by agar diffusion tests. Their antibacterial activity is thus determined by the distance of the growth inhibition zone (the recorded diameter). It should be noted here that the Zn content of GR-MOF-8 and 9 is relatively high (*ca.* 29 wt%) in comparison with other MOFs (MOF-5 with 18.3 wt% or TMU-3 with 10.5 wt%) [31], which could be a priori associated with a stronger effect.

*SA* and *EC* are considered to be two of the most relevant human pathogens, and some of their strains cause a variety of infections to wild animals, pets and livestock [32,33,34]. Moreover, their treatment is suggested by the development of resistances to each new class of antimicrobial drug, which sustain the pursuit of more efficient bactericidal therapies [35].

As shown in Figure 3, GR-MOF-8 exhibited the highest inhibitory action against both bacteria strains, being particularly evidenced towards *SA*, with an increasing inhibition effect over 14 days of contact time. A large inhibition zone generally corresponds to lower minimal inhibitory concentrations (*MICs*) because less amount of compound is needed to achieve the same antibacterial effect [36]. This effect could be ascribed to its highest release and diffusion of Zn^2+^ ions into the agar plates. In other words, the balance between material stability and its release capacity, *e.g.,* large inhibition profile, is generally correlated as an inverse linear relationship, where higher the active zone, the lower the antibacterial agent concentration [36].

In this sense, the role of each precursor (the ligand H_3_L and the metal salt Zn(AcO)_2_·2H_2_O) on the bacterial inhibition of GR-MOF-8 and 9 were investigated (Appendix A). In particular, the H_3_L linker exclusively did not produce any significant inhibition zones against *EC* and *SA*, whereas a similar antimicrobial performance was observed by the MOF equivalent amount of metal ion in the Zn(AcO)_2_·2H_2_O control, in comparison with the whole MOFs. Considering the progressive degradation of MOFs over time (Appendix A), this antibacterial activity profile could be explained due to the slower diffusion rate of the Zn^2+^ ion release from GR-MOFs. Furthermore, when comparing the antibacterial activity of both materials, with similar Zn content (27.94 and 28.74 %w for GR-MOF-8 and 9, respectively), one could suggest that the progressive antibacterial activity of GR-MOF-8 is directly related to its slow degradation rate. Thus, GR-MOF-8 can act as a reservoir of metal ions providing a gradual release and resulting in a sustained antibacterial action.

## 4. Conclusions

Two novel bioactive MOF materials (denoted GR-MOF-8 and 9) based on Zn^2+^ and the 5-((4-carboxyphenyl)ethynyl) isophthalate linker have been successfully synthetized and their structures fully characterized. Both materials are 3D structures, displaying pores (21.0 × 10.9 Å, and 18 Å for GR-MOF-8 and 9, respectively) along the *a* crystallographic axis, as confirmed by single crystal X-ray diffractometry.

Remarkably, both materials show an antibacterial activity against *EC* and *SA*, maintained over time (GR-MOF-8), which is likely due to the progressive release of Zn, acting as reservoirs. These novel materials offer interesting possibilities in antibacterial surface development (self-cleaning), or prostheses.

## Figures and Tables

**Figure 1 nanomaterials-12-04139-f001:**
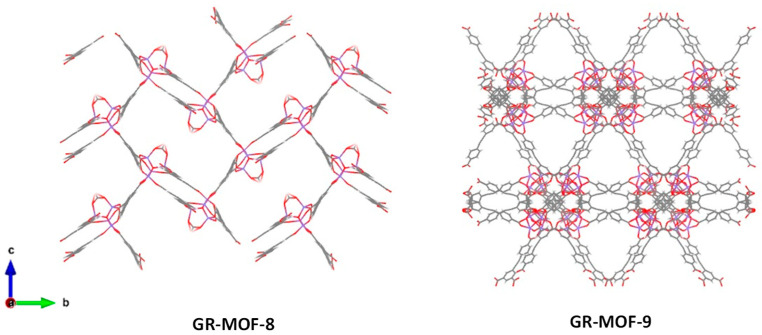
Three-dimensional representation of GR-MOF-8 and 9 along *a* axis. Zn, C, O and H are represented in blue, grey, red and white, respectively.

**Figure 2 nanomaterials-12-04139-f002:**
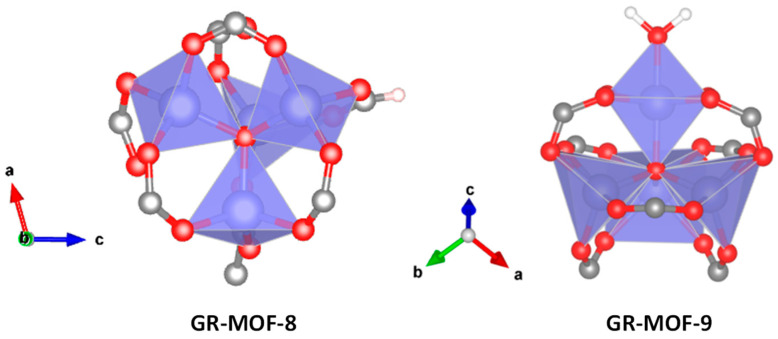
Schematic view of the Zn tetra-tetrahedra nodes of GR-MOF-8 and 9. Zn, C, O and H are represented in blue, grey, red and white, respectively.

**Figure 3 nanomaterials-12-04139-f003:**
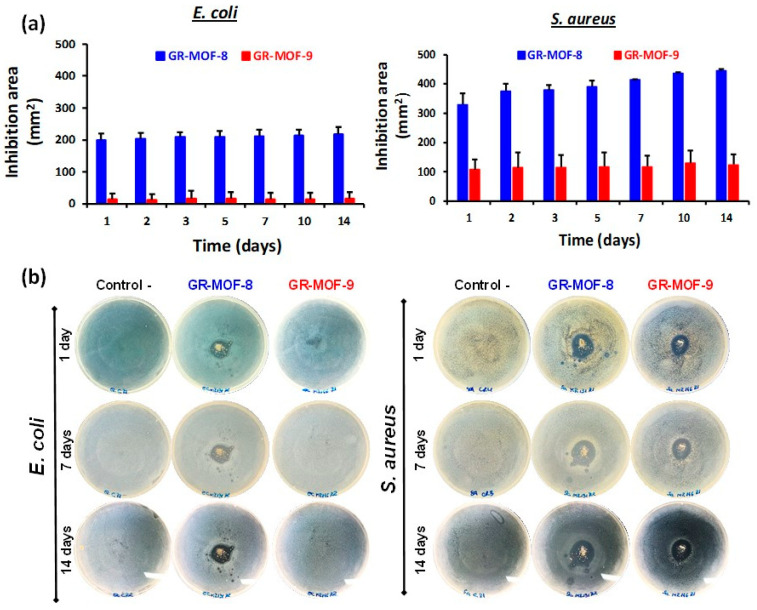
(**a**) Halo inhibition zone (expressed in mm^2^) for GR-MOF-8 (blue) and GR-MOF-9 (red) along 14 days with EC (**left**) and SA culture (**right**) in agar plates at 37 °C. Error bars represent standard deviation; (**b**) representative images of inhibition experiments with EC and SA culture corresponding to GR.MOF-8 and 9 after diverse contact times.

**Table 1 nanomaterials-12-04139-t001:** Crystallographic data and structure refinement details of compounds GR-MOF-8 and 9.

	GR-MOF-8	GR-MOF-9
**Formula**	C_35_H_14_O_15_Zn_4_	C_34_H_16_O_14_Zn_4_
**M_r_**	935.94	909.95
**Crystal system**	Monoclinic	Tetragonal
**Space group (no.)**	*P2_1_/c*	*I-4*
** *a* ** **(Å)**	16.3308(17)	22.1989(13)
** *b* ** **(Å)**	31.267(3)	22.1989(13)
** *c* ** **(Å)**	16.7781(17)	42.516(3)
** *α* ** **(°)**	90	90
** *β* ** **(°)**	106.443(3)	90
** *γ* ** **(°)**	90	90
**V (Å^3^)**	8216.8(14)	20,952(3)
**Z**	4	8
**T (K)**	150(2)	200(2)
***ρ*_calc_ (g/cm^3^)**	0.757	0.577
**μ (mm^−1^)**	1.185	0.928
**F(000)**	1856	3616
**Radiation**	MoKα (0.71073 λ)	MoKα (0.71073 λ)
**Index ranges**	−13 ≤ h ≤ 13, −26 ≤ k ≤ 26, −13 ≤ l ≤ 13	−18 ≤ h ≤ 18, −18 ≤ k ≤ 18, −35 ≤ l ≤ 35
**GoF on F^2^**	1.073	2.541
**Final R indices (I >= 2σ (I))**	R_1_ = 0.0776; wR_2_ = 0.2169	R_1_ = 0.1753; wR_2_ = 0.4279
**Final R indices (all data)**	R_1_ = 0.0926; wR_2_ = 0.2304	R_1_ = 0.2030; wR_2_ = 0.4922

## Data Availability

Data Availability Statements in section “MDPI Research Data Policies” at https://www.mdpi.com/ethics, 15 October 2022.

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
