# Peer review of "Antibacterial Activity of Two Zn-MOFs Containing a Tricarboxylate Linker"

_nanomaterials, 2022, doi:10.3390/nano12234139_

Round 1
Reviewer 1 Report
In this paper, the author reported two novel Zn-based MOFs, GR-MOF-8 and GR-MOF-9. Its structure was determined by single crystal X-ray diffraction (SCXRD) and was fully characterized by Fourier transform infrared (FTIR), elemental analysis and thermogravimetry. Both materials show certain antibacterial activity against EC and SA. However, the author only conducted a simple bacteriostatic experiment on these two cases of Zn MOFs. The main highlight of it is the antibacterial activity, while there is not enough experimental data to support the conclusion. Detailed comments are listed below.
Comment 1: The title of the article is "Zn-MOF reserves with long-term bacterial effect", but the author provides the experimental results of inhibition. The experimental content is inconsistent with the article title.
Comment 2: In the process of antibacterial, in what way is Zn2+ ion released from Zn-MOF? The author should clearly put forward what method to prove the release of Zn2+ ions. The author should provide the release curve of Zn2+ ion in the inhibition experiment of GR-MOF-8 and GR-MOF-9.
Comment 3: In section 3.3 "Antibacterial activity", what is the specific value of "minimum inhibitory concentration (MIC)"? The author should list it clearly.
Comment 4: In the Figure 3a (right), the error value is too large, and the author should consider whether the experimental data is reliable.
Comment 5: In the paper, the author should use experimental data to prove the antibacterial effect of GR-MOF-8&9. What is the purpose of selecting Zn(AcO)2×2H2O for inhibition experiment comparison? The author should choose to compare it with Zn-MOF, such as ZIF-8.
6. more systemetic literature review should be done and some new MOF literatures should be cited.
Author Response
To facilitate reading, we have indicated in bold text the remarks of the referees, in black our answers and in red the modified text in the manuscript.
Reviewer 1
In this paper, the author reported two novel Zn-based MOFs, GR-MOF-8 and GR-MOF-9. Its structure was determined by single crystal X-ray diffraction (SCXRD) and was fully characterized by Fourier transform infrared (FTIR), elemental analysis and thermogravimetry. Both materials show certain antibacterial activity against EC and SA. However, the author only conducted a simple bacteriostatic experiment on these two cases of Zn MOFs. The main highlight of it is the antibacterial activity, while there is not enough experimental data to support the conclusion. Detailed comments are listed below.
- Comment 1: The title of the article is "Zn-MOF reserves with long-term bacterial effect", but the author provides the experimental results of inhibition. The experimental content is inconsistent with the article title.
We thank the reviewer for this suggestion. As requested, we have decided to change the title by “Antibacterial activity of two Zn-MOFs containing a tricarboxylate linker“. We have just focused on the antibacterial activity of these novel Zn-MOFs as proof of concept.
- Comment 2: In the process of antibacterial, in what way is Zn2+ ion released from Zn-MOF? The author should clearly put forward what method to prove the release of Zn2+ions. The author should provide the release curve of Zn2+ion in the inhibition experiment of GR-MOF-8 and GR-MOF-9.
Regarding the release of Zn2+ ion of the reviewer, we have performed a novel assay evaluating the stability of this material. The direct performance in presence of water &/or both media has been ruled out due to the instant dissolution of Zn-MOF under these conditions and the complexity of this media (density, viscosity, etc.), which hamper the progressive analysis of its degradation profile. Therefore, the structural stability has been monitored under controlled relative humidity (RH 70%), for simulating bacteria culturing conditions at different time points.
After 72 h (Figure S8), the structural stability is maintained, although a particle loss of the crystallinity coupled with the presence of new diffraction peaks is observed. This data supports the idea of a gradual MOF degradation, and therefore, a gradual Zn2+ release.
This new data has been added in the SI and commented in the main text.
Considering the progressive degradation of MOFs over time (Figure S8), this antibacterial activity profile could be explained due to the slower diffusion rate of the Zn2+ ion release from GR-MOFs. Further, when comparing the antibacterial activity of both materials, with similar Zn content (27.94 and 28.74 %w for GR-MOF-8 & 9, respectively), one could suggest that the progressive antibacterial activity of GR-MOF-8 is directly related with its slow degradation rate. Thus, the GR-MOF-8 can act as a reservoir of metal ions providing a gradual release and resulting in a sustained antibacterial action.
- Comment 3: In section 3.3 "Antibacterial activity", what is the specific value of "minimum inhibitory concentration (MIC)"? The author should list it clearly.
We appreciate this remark about the MIC values. In this line, as the first proof of concept, the antibacterial activity for these novel Zn-MOFs has been investigated using a specific concentration (5 mg) in order to verify qualitatively their potential effect, thus, it renders the possibility to calculate their MIC value. This particular assay will be crucial in the near future for a deeply investigation of the antibacterial activity of these new MOFs.
However, we have now included a sentence and a references in order to facilitate the readers to understand the relationship between the agent concentration with its potential effects.
(3.3. Antibacterial Activity section, page 7): “This effect could be ascribed to its highest release and diffusion of Zn2+ ions into the agar plates. In other words, the balance between material stability and its release capacity: e.g., large inhibition profile is generally correlated as an inverse linear relationship, where higher the active zone, lower is the antibacterial agent concentration.[35]”
- Comment 4: In the Figure 3a (right), the error value is too large, and the author should consider whether the experimental data is reliable.
We are aware about the error value of the halo inhibition. Despite the observed error (mainly for GR-MOF-9), it seems to be sustained over the bacterial contact time (since day 1 to day 14), which could be related with a potential deposition variability (e.g., more electrostatic material, different behavior with the broth media, etc.). Through this assay, we were able to exhibit the bacterial effect induced by this new MOF, demonstrating its active impact. Thus, even if it´s more qualitative, we consider these data to be significant and relevant.
- Comment 5: In the paper, the author should use experimental data to prove the antibacterial effect of GR-MOF-8&9. What is the purpose of selecting Zn(AcO)2×2H2O for inhibition experiment comparison? The author should choose to compare it with Zn-MOF, such as ZIF-8.
We partially disagree with the Referee in this point. The selection of the Zn(AcO)2×2H2O is due to its involvement for the production of GR-MOF-8 & 9 as the main metal precursor along with its recognized antibacterial activity. MOFs are well-known for transporting and delivering diverse active ingredients (e.g., pestices, drugs, aminoacids, etc.; See Rojas et al., Chem.Rev. 2019, Giménez-Marqués et al., Coord.Chem.Rev, 2016.) as well as to introduce into the MOF framework, constituents with additional properties Thus, the Zn+2 effect and its comparison with the whole MOF structure will allow us to shed light on the mechanism of action. As observed, the linker did not induce any type of activity, being so mainly produced by the metal.
On the other hand, the suggestion to use experimentally other MOFs as control was not considered priority. Only few examples of Zn-based MOFs with antibacterial activity have been reported. In line with comment 5 and 6 we have added a new example. So, in this paper we are mentioning the most relevant works on the antibacterial activity of Zn-based MOFs.
- Comment 6. More systematic literature review should be done and some new MOF literatures should be cited.
We appreciate this remark. We have now revised and modified the main text, adding an newly reported study.
In particular, Zn-based MOFs can be seen as promising reservoir materials of metal ions with antimicrobial activity.[19] However, so far, only few examples of Zn-based MOFs with antibacterial activity have been reported. Wang et al., reported a series of four MOFs based on Zn2+ and 2,2’-bipyridine-4,4’-dicarboxylic acid with different morphologies and architectures (1D nanobelts, 3D nanorod-flowers, fan-like and rhombus-like).[20] The antimicrobial studies against Bacillus subtilis, Staphylococcus aureus, Salmonella enteritidis, Escherichia coli, Proteus vulgaris and Pseudomonas aeruginosa demonstrated the influence of the MOF particles’ morphology on their activity, particularly that fan-like particles are more biocides against gram-negative bacteria than other morphologies. A further study reported the possible preparation of a hydrogel containing a previously reported Zn-based MOF with antibacterial activity against S. aureus and E. coli.[21] Some of us described the MOF material BioMIL-5, based on Zn2+ and the natural dicarboxylic acid, azelaic acid.[22] When tested against S. aureus and Staphylococcus epidermidis, BioMIL-5 demonstrated that the antimicrobial activity of the individual components of BioMIL-5 were maintained after its synthesis. Other recent example is the effect of the media in the antibacterial activity of the Zn-based MOF ZIF-8, with superior antibacterial activity against E. coli in phosphate buffer saline (PBS) of Luria Bertani (LB) compared with pure water.[23]

Reviewer 2 Report
Rojas et al. designed a Zn based MOF for bactericidal application. They synthesized two new MOFs, which were characterized by SCXRD. However, the reviewer found several major issues need to be profoundly revised,
1. The reviewer is suggesting the author provide CIF file for helping evaluate the SCXRD result.
2. The authors should clearly perform a control experiment to know whether the MOF is still stable inside agar plates. Meanwhile, the use of zinc salt instead of MOF can lead to a more efficient bacteria inhibition effect. The author should discuss the potential advantage of using MOF for bactericidal application. There is no doubt the use of MOF is quite expensive considering the high cost of H3L.
3. The different bactericidal effects between GR-MOF-8 and GR-MOF-9 should be studied more in detail to provide useful information for the rational design new MOF with a better bactericidal effect.
Author Response
To facilitate reading, we have indicated in bold text the remarks of the referees, in black our answers and in red the modified text in the manuscript.
Reviewer 2
Rojas et al. designed a Zn based MOF for bactericidal application. They synthesized two new MOFs, which were characterized by SCXRD. However, the reviewer found several major issues need to be profoundly revised,
- The reviewer is suggesting the author provide CIF file for helping evaluate the SCXRD result.
We thanks for this suggestion. We have now added the CIF file in order to facilitate the evaluation of the SCXRD of GR-MOF-8 & 9.
- The authors should clearly perform a control experiment to know whether the MOF is still stable inside agar plates. Meanwhile, the use of zinc salt instead of MOF can lead to a more efficient bacteria inhibition effect. The author should discuss the potential advantage of using MOF for bactericidal application. There is no doubt the use of MOF is quite expensive considering the high cost of H3L.
We appreciate this Referee´remarks. The antibacterial performance was prepared with agar diffusion following standard protocols plate, where the whole MOF was in direct contact with the media, as proof of concept. The MOF inclusion into the agar (before its solidification) will be other experimental approach that could be interested in the next future, in case its needed a stronger effect. The main aim here was to demonstrate for the first time if both MOF possess any type of antibacterial activity, as stated in the manuscript.
On the other hand, the advantage to use MOFs for bacterial applications is because they can act as promising reservoirs of the active metal, in this case just the Zn2+, prolonging its release over the time, as observed up to 14 days. Moreover, the characteristic porosity of MOFs allows them to carry other type of active ingredients (e.g., antibacterial drugs such as gentamicin, isoniazid, etc.), which will boost the antibacterial therapy with a potential dual effect.
It is true that the synthesis of the H3L linker is quite expensive. So, this study should be considered as a proof-of-concept article that describes for the first time two novel MOF structure with some interesting activity (antibacterial).
- The different bactericidal effects between GR-MOF-8 and GR-MOF-9 should be studied more in detail to provide useful information for the rational design new MOF with a better bactericidal effect.
We are grateful for this suggestion, and we agree with the Reviewer point of view. This investigation was focused to point out the potential antibacterial activity of these original and new metal-organic frameworks, GR-MOF-8 & 9. Therefore, as proof of concept, the experimental design was addressed in the best working conditions for simulating the bacteria culturing conditions. These generated outcomes will support the next investigations in order to provide more deeply information about their performance and potential industrial translation.

Reviewer 3 Report
The article is devoted to a relevant topic with great practical potential. The article is well-written, in accordance with all the requirements of journal. Two remarks I would note are the elemental analysis for GR-MOF-8: Elemental analysis (weight, %); Calculated: C, 47.15; H, 3.22; N, 3.84; Found: C, 42.12; H, 4.85; N, 3.28 (Lines 104-105). Analisys should be improved; And elemental analysis for GR-MOF-9: Calculated: C, 47.15; H, 3.22; N, 3.84; Found: C, 47.15; H, 3.22; N, 3.84 (Lines 112-113). Analysis needs to be checked.
Overall, I am impressed with the article and believe that it can be accepted after minor revision.

Author Response
To facilitate reading, we have indicated in bold text the remarks of the referees, in black our answers and in red the modified text in the manuscript.
Reviewer 3
The article is devoted to a relevant topic with great practical potential. The article is well-written, in accordance with all the requirements of journal. Two remarks I would note are the elemental analysis for GR-MOF-8: Elemental analysis (weight, %); Calculated: C, 47.15; H, 3.22; N, 3.84; Found: C, 42.12; H, 4.85; N, 3.28 (Lines 104-105). Analisys should be improved; And elemental analysis for GR-MOF-9: Calculated: C, 47.15; H, 3.22; N, 3.84; Found: C, 47.15; H, 3.22; N, 3.84 (Lines 112-113). Analysis needs to be checked.
Overall, I am impressed with the article and believe that it can be accepted after minor revision.
We appreciate the positive comments of the Referee about our work and acceptance for the potential submission at Nanomaterials journal.

Round 2
Reviewer 1 Report
this work can be published
Reviewer 2 Report
The authors have addressed all the comments I raised. I recommend accepting this paper.